# Personalized Moderate-Intensity Exercise Training Combined with High-Intensity Interval Training Enhances Training Responsiveness

**DOI:** 10.3390/ijerph16122088

**Published:** 2019-06-13

**Authors:** Bryant R. Byrd, Jamie Keith, Shawn M. Keeling, Ryan M. Weatherwax, Paul B. Nolan, Joyce S. Ramos, Lance C. Dalleck

**Affiliations:** 1Recreation, Exercise & Sport Science Department, Western Colorado University, Gunnison, CO 81231, USA; bryant.byrd@western.edu (B.R.B.); jamiemichellekeith@gmail.com (J.K.); skeeling@western.edu (S.M.K.); rweatherwax@western.edu (R.M.W.); 2SHAPE Research Centre, Exercise Science and Clinical Exercise Physiology, College of Nursing and Health Sciences, Flinders University, Adelaide, SA 5001, Australia; paul.nolan@flinders.edu.au (P.B.N.); joyce.ramos@flinders.edu.au (J.S.R.); 3Centre for Research on Exercise, Physical Activity and Health, School of Human Movement and Nutrition Sciences, The University of Queensland, Brisbane, QLD 4072, Australia

**Keywords:** cardiometabolic risk factor, metabolic syndrome, responders, translational research

## Abstract

This study sought to determine if personalized moderate-intensity continuous exercise training (MICT) combined with high-intensity interval training (HIIT) was more effective at improving comprehensive training responsiveness than MICT alone. Apparently healthy, but physically inactive men and women (*n* = 54) were randomized to a non-exercise control group or one of two 13-week exercise training groups: (1) a personalized MICT + HIIT aerobic and resistance training program based on the American Council on Exercise guidelines, or (2) a standardized MICT aerobic and resistance training program designed according to current American College of Sports Medicine guidelines. Mean changes in maximal oxygen uptake (VO_2max_) and Metabolic (MetS) z-score in the personalized MICT + HIIT group were more favorable (*p* < 0.05) when compared to both the standardized MICT and control groups. Additionally, on the individual level, there were positive improvements in VO_2max_ (Δ > 4.9%) and MetS z-score (Δ ≤ −0.48) in 100% (16/16) of participants in the personalized MICT + HIIT group. In the present study, a personalized exercise prescription combining MICT + HIIT in conjunction with resistance training elicited greater improvements in VO_2max_, MetS z-score reductions, and diminished inter-individual variation in VO_2max_ and cardiometabolic training responses when compared to standardized MICT.

## 1. Introduction

It is well established that a low cardiorespiratory fitness (CRF) level is a risk factor for coronary heart disease and cardiovascular disease (CVD) mortality [1]. Regular aerobic exercise training following standardized guidelines has been demonstrated to be an effective tool in improving CRF [2]. However, not all individuals respond positively to such exercise. Indeed, there is considerable individual variability in training adaptations including the so-termed ‘non-responders’ and, in some instances, ‘adverse responders’ [3,4]. This variability in training responsiveness is not well understood and may be attributable to various factors including the absence of set definitions in the literature for responders/non-responders and a one size fits all approach to exercise prescription [5].

It has been purported that a more ‘individualized/personalized approach’ to the exercise prescription may improve overall training responsiveness [6]. Indeed, it has been previously demonstrated that an ‘exercise intensity’ prescription that is tailored according to a threshold-based model (i.e., ventilatory thresholds) results in 100% of participants improving CRF relative to only about 42% when exercise intensity is ‘standardized’ or prescribed according to a relative percent method (i.e., % heart rate (HRR)) [7]. Our most recent study also suggests that a personalized approach to exercise prescription compared with a standardized protocol may also induce better improvement in the clustering of cardiovascular risk factors, depicted as a reduction in metabolic syndrome (MetS) severity [8]. The variability in CRF and cardiometabolic response to a standardized exercise prescription has been put forth to be a result of its inability to account for individual metabolic differences [9].

The paradigm of high-intensity interval training (HIIT) provides an alternative time-efficient exercise strategy for public health promotion. Indeed, relative to moderate intensity continuous exercise training (MICT), HIIT has been demonstrated to elicit comparable or superior improvements in CRF [10] and MetS [11,12], despite a lesser time commitment. However, the American College of Sports Medicine (ACSM) recommends HIIT following an initial conditioning phase (typically 2–3 months) and only on a limited basis to avoid excessive orthopedic stress [13]. Moreover, HIIT may briefly elevate the risk of cardiac events in persons with underlying undiagnosed CVD and thus initial supervision for this type of exercise may be required in untrained and high-risk individuals. Therefore, it seems practical and consistent with current physical activity recommendations to combine the two types of training (MICT + HIIT) to increase the likelihood of positive cardiovascular and metabolic health adaptations, while simultaneously maintaining safety [14].

The purpose of this study was to determine if personalized MICT combined with HIIT would be more effective at improving training responsiveness than MICT alone. This question was addressed by assessing inter-individual variation in CRF and creating a MetS z-score to assess cardiometabolic training responsiveness. It was hypothesized that personalized MICT combined with HIIT would be more effective at improving comprehensive training responsiveness to CRF and cardiometabolic improvement when compared to MICT alone.

## 2. Materials and Methods

Fifty-four nonsmoking men and women (21 to 55 years) were recruited from a local university and surrounding community via advertisement through the university website, local community newspaper, and word-of-mouth. Participants were eligible for inclusion in the study if they were low-to-moderate risk as defined by the American College of Sports Medicine and not physically active [13]. This study was approved by the Human Research Committee at Western Colorado University (HRC2017-02-03-R28). All participants provided informed consent in advance of their participation in the study.

### 2.1. Experimental Design

Participants were randomized to a non-exercise control group (who were instructed to continue their usual lifestyle habits) or one of two exercise training groups and subsequently completed baseline testing (Figure 1). Participants randomized to the exercise training groups performed 13 weeks of exercise training according to one of two programs: (1) a personalized MICT + HIIT exercise program based on the American Council on Exercise (ACE) Integrated Fitness Training (IFT) model guidelines [15], or (2) a standardized MICT program designed according to current ACSM guidelines [13]. Each exercise training group performed a similar frequency and duration of exercise training. Participants within all three groups completed post-program testing after the 13-week intervention. Assessments of anthropometric measures, cardiometabolic risk factors, and maximal oxygen uptake (VO_2max_) were obtained in duplicate at baseline in order to develop personalized criteria for the identification of responders and non-responders. There was also a duplicate measure of VO_2max_ at post-program.

### 2.2. Moderate-Intensity Continuous Training (MICT)

The MICT for both exercise groups was performed on various aerobic modalities: cycle and rowing ergometers, elliptical crosstrainer, and treadmill. The exercise intensity method for MICT differed between treatment groups. The standardized group was prescribed exercise intensity for MICT according to a percentage of HRR. Conversely, the personalized group was prescribed exercise intensity for MICT according to ventilatory thresholds. In both exercise training groups, a target heart rate (HR) coinciding with either the prescribed HRR zone or prescribed threshold zone (Figure 1) was used to establish a specific exercise training intensity for each exercise session of MICT. In the personalized group target HR for MICT was established in the following manner:**Wk 1–4 (HR < VT1):** target HR = HR range of 10–15 bpm just below VT1**Wk 5–8 (HR ≥ VT1 to < VT2):** target HR = HR range of 10–20 bpm (above VT1 and below VT2)**Wk 9–13 (HR ≥ VT2):** target HR = HR range of 10–15 bpm at or just above VT2

The exercise prescription for MICT was progressed according to recommendations made elsewhere by the ACE [15] and ACSM [13]. Polar HR monitors (Polar Electro Inc., Woodbury, NY, USA) were used by trained researchers to monitor the HR response during all exercise sessions. Researchers adjusted workloads on aerobic modalities accordingly during each MICT exercise session to ensure that actual HR responses aligned with the target HR. All MICT exercise prescription details for each group over the course of the 13-week training period are presented in Figure 1.

### 2.3. High-Intensity Interval Training (HIIT)

The personalized group interspersed MICT each week with one session of HIIT. All HIIT sessions were performed on the treadmill and commenced with a 5 min warm up performed at a light intensity (30–<40% HRR) and concluded with a 5 min cool-down also completed at light intensity [13]. The HIIT protocol involved eight, 60 s interval bouts performed at the workload corresponding to 100% VO_2max_, separated by 150 s active recovery bouts performed at light intensity (30–<40%HRR). The HIIT protocol was based on previous research from our group [15]. After four weeks of HIIT, the number of interval bouts increased from eight to 10 for the next four weeks of training (week 5–8). For the last five weeks (week 8–13), the number of interval bouts increased to 12. The active recovery duration was held constant at 150 s throughout the 13-week intervention. All HIIT exercise prescription details for the personalized training group over the course of the 13-week training period are presented in Figure 1.

### 2.4. Resistance Exercise Prescription

Resistance training commenced during week four of the overall study for both treatment groups and was subsequently completed 3 days a week for the remainder of the intervention. All sessions were supervised by researchers who closely monitored adherence to the prescribed program, ensured proper technique for each exercise, and provided specific information on progression.

### 2.5. Resistance Training for the Standardized Group

The resistance training program for the standardized treatment group was designed according to ACSM guidelines [13] and consisted of single and multi-joint exercises completed using machine modalities. The following exercises were performed: bench press, shoulder press, lateral pulldown, seated row, bicep curl, tricep pushdown, seated leg press, seated leg extension, prone lying leg curl, and seated back extension/flexion. Resistance training for all sessions consisted of two sets of 12 repetitions. Each exercise was completed at a moderate intensity of 5–6 on the modified Borg rating of perceived exertion (RPE) scale [16] and rated according to guidelines published by Sweet and colleagues [17]. Resistance was progressed every 2 weeks by ~3–5% of total weight lifted for the upper body and ~6–10% for lower-body exercises so that the session RPE of 5–6 was maintained across the training program.

### 2.6. Resistance Training for the Personalized Group

The resistance training program for participants in the personalized treatment group was designed according to ACE guidelines [15] and consisted of multijoint/multiplanar exercises completed using free weight and machine modalities. The machine modalities that were used allowed for free motion during the exercise and therefore the range of motion was not limited to a specific arc. The following exercises were performed in the personalized treatment group: stability ball circuit (hip bridges, crunches, Russian twists, planks), lunge matrix, kneeling/standing wood chops, kneeling/standing hay bailers, dumbbell squat to 90-degree knee bend, standing one-arm cable row, step-ups with dumbbell onto 15cm step, modified (assisted) pull-ups, and dumbbell bench press. Resistance training for all sessions consisted of two sets of 12 repetitions. The intensity of weighted exercises started at 50% 5-RM and was progressed by 5% 5-RM increments every 2 weeks. For exercises that did not include a weighted resistance (e.g., stability ball circuit, modified pull-ups), the volume of each exercise in the form of repetitions was increased by ~5–10% to maintain an RPE rating of 5–6.

### 2.7. Anthropometric, Cardiovascular, and Cardiometabolic Measurements

Various anthropometric, cardiovascular, and cardiometabolic measurements were obtained for all participants at baseline and 13 weeks. Anthropometric measurements included height, weight, waist circumference, and percent body fat. Cardiovascular measurements were comprised of resting heart rate and blood pressure. Cardiometabolic measurements consisted of fasting blood lipids and glucose. The procedures for each of these assessments were consistent with our previous research and detailed elsewhere [7].

### 2.8. Metabolic Syndrome z-Score

A continuous risk score assessment scale (MetS z-score) has been used previously by our research group and others to identify changes in metabolic syndrome (MetS) severity following MICT and HIIT interventions [5,12]. The MetS severity was presented as a sex-specific MetS z-score calculated using the following equations [18]: (1) MetS z-score_men_ = [(40 − HDL-C)/8 × 9] + [(TG − 150/69)] + [(FG − 100)/17 × 8] + [(WC − 102)/11 × 5] + [(MAP − 100)/10 × 1]; (2) MetS z-score_women_ = [(50 − HDL-C)/14 × 5] + [(TG − 150/69)] + [(FG − 100)/17 × 8] + [(WC − 88)/12 × 5] + [(MAP − 100)/10 × 1], where FG = fasting glucose; HDL-C = high-density lipoprotein cholesterol; MAP = mean arterial pressure; TG = triglycerides; and WC = waist circumference.

### 2.9. Muscular Fitness Assessments

The procedures for the assessment of muscular fitness outlined elsewhere were followed [15]. Participants performed five-repetition maximum (5-RM) testing for the bench press and leg press exercises to assess muscular fitness. The following protocol was used for 5-RM testing:10 repetitions of a weight the participant felt comfortable lifting (40–60% of estimated 5-RM) were performed to warm up muscles followed by 1 min rest period5 repetitions at a weight of 60–80% estimated 5-RM was performed as a further warm up and followed by a 2 min rest periodFirst 5-RM attempt at a weight of 2.5–20 kg greater than warm up
If the first 5-RM lift was deemed successful by the researcher (appropriate lifting form) weight was increased until maximum weight participant can lift was established with 3 min between each attempt.If the first 5-RM lift deemed unsuccessful by the researcher, weight was decreased until the participant successfully lifted the heaviest weight possible.

There were 3 min rest between 5-RM attempts and a maximum of 3 × 5-RM attempts. There was 5 min of rest between the 5-RM testing of each resistance exercise.

### 2.10. Maximal Exercise Testing and Verification Bout Trial Procedures

Participants completed a maximal graded exercise test (GXT) according to our previously described protocol [7]. Gas exchange data were averaged for every 15 s and VO_2max_ was determined by averaging the final two valid 15 s VO_2_ samples. The highest achieved HR during the GXT was considered the maximal HR (HR_max_). In order to confirm ‘true’ VO_2max_, a verification trial was performed 20 min following the completion of the GXT trial. The protocol for the verification trial included a 4 min warm up followed by a volitional test to exhaustion at a constant workload that was set at 5% higher than the last completed stage of the GXT. Gas exchange data and HR were continuously monitored and averaged every 15 s. The VO_2max_ during the verification trial was established in the same manner at the GXT trial, with the average of the final two valid 15 s samples for VO_2_. A ‘true’ VO_2max_ was confirmed if the GXT trial and verification trial VO_2max_ values were within ± 3.0%, as used previously by our group [19]. Subsequently, participant ‘true’ VO_2max_ value was considered to be a mean of the GXT trial VO_2max_ and verification trial VO_2max_. If there was a greater than ± 3.0% difference in VO_2max_ values between the GXT and verification bout trials, participants repeated the GXT with verification trial protocol within 24–72 h until ‘true’ VO_2max_’ was confirmed with the ± 3.0% criterion.

### 2.11. Quantification of Ventilatory Thresholds

Similar to our past research [7], VT1 and VT2 were quantified by visual inspection of graphs of time plotted against each relevant respiratory variable (according to 15 s time-averaging). The criterion for VT1 was an increase in Ventilation (VE)/VO_2_ without a concomitant rise in VE/VCO_2_ along with a departure from the linearity of VE. The criterion for VT2 was a concurrent increase in both VE/VO_2_ and VE/VCO_2_ [20]. The quantification of all ventilatory threshold measurements was performed by two PhD-trained exercise physiologists with 35 years of collective experience.

### 2.12. Statistical Analyses

All analyses were performed using SPSS Version 24.0 (IBM Corporation, New York, NY, USA) and GraphPad Prism 7.0. (San Diego, CA, USA). The sample size was determined from Skinner and colleagues [21], assuming a power of 0.90 and an effect size of 0.8. Therefore, 15 subjects would be needed for each of the three groups (total *n* required = 45) [22]. However, we assumed there would be an approximate 20% dropout rate based on previous exercise training studies [23] and recruited and randomized an additional three participants to each of the exercise training groups and non-exercise control group.

Measures of centrality and spread are presented as mean ± standard deviation (SD). All baseline-dependent variables were compared using a general linear model (GLM) ANOVA and, where appropriate, Tukey post hoc tests. Within-group comparisons were made using paired t-tests. Other between-group 13-week changes were analyzed using GLM-ANOVA and, where appropriate, Tukey post hoc tests. The between-group difference of MetS z-score change following the exercise intervention was assessed through ANCOVA, with sex, age, and baseline values assigned as covariates. The assumption of normality was tested by examining normal plots of the residuals in ANOVA models. Residuals were regarded as normally distributed if Shapiro–Wilk tests were not significant [22].

The coefficient of variation (CV) for VO_2max_ was calculated with the duplicate baseline measures as previously described [24]. The calculated CV values were used to determine site-specific technical error (biological variability + measurement error) values for VO_2max_ [25,26]. To determine individual VO_2max_ training responsiveness delta values (Δ) were calculated (post-program minus baseline value) to establish the percent change (%Δ = Δ/baseline VO_2max_) in VO_2max_. The %Δ in VO_2max_ values were compared to the calculated site-specific technical error (i.e., %Δ > CV). Subsequently, participants were categorized as a responder if their %Δ was greater than the site-specific technical error criterion or a non-responder if the %Δ failed to exceed the site-specific criterion. The calculated site-specific technical error for VO_2max_ equated to %Δ > 4.9%.

The CV for MetS z-score was calculated with the duplicate cardiometabolic baseline measures to quantify the site-specific technical error (biological variability + measurement error) for MetS z-score. To determine individual MetS z-score training responsiveness delta values (Δ) were calculated (post-program minus baseline value) to establish the change (Δ) in MetS z-score. The Δ in MetS z-score was compared to the calculated site-specific technical error (i.e., MetS z-score CV). Subsequently, participants were categorized as a responder if their Δ was greater than the site-specific technical error criterion or a non-responder if the Δ failed to exceed the site-specific criterion. The calculated site-specific technical error for MetS z-score equated to Δ > −0.48.

Chi-square (*χ*^2^) tests were used to analyze the incidence of responders and non-responders for VO_2max_ and MetS z-score following the intervention separated by treatment group (standardized MICT and personalized MICT + HIIT) with a subsequent Cramer’s V test to quantify effect size. The probability of making a Type I error was set at *p* < 0.05 for all statistical analyses.

## 3. Results

All analyses and data presented in the results are for those participants who completed the investigation. The physical and physiological characteristics for participants at baseline and 13 weeks are presented in Table 1. At baseline, treatment (standardized MICT and personalized MICT + HIIT) and non-exercise control groups did not differ significantly in physical or physiological characteristics (all *p* > 0.05) with the exception of blood glucose values (*p* < 0.05).

The exercise prescriptions in both treatment groups were well-tolerated for the 32 of 36 participants who completed the study. Four participants were unable to complete the study for the following reasons: illness (*n* = 2) and personal reasons (*n* = 2). Dropout was similar in both treatment groups. Overall, there was excellent adherence to the total number of prescribed training sessions: standardized MICT group – mean, 90.3% (range, 81.7–100%) and personalized MICT + HIIT group – mean, 91.0% (range, 80–100%).

After 13 weeks, mean changes in VO_2max_ and MetS z-score in the personalized MICT + HIIT group were significantly more favorable (*p* < 0.05) when compared to both the standardized MICT and control groups. Changes in percent body fat and waist circumference from baseline to 13 weeks in the standardized MICT and personalized MICT + HIIT groups were also significantly greater (*p* < 0.05) when compared with the control group.

### 3.1. Incidence of VO_2max_ Responders and Non-Responders

The incidence of VO_2max_ responders and non-responders to exercise training in both the standardized MICT and personalized MICT + HIIT groups are shown in Figure 2. In the standardized MICT group 68.75% (11/16) of individuals were categorized as responders (change in VO_2max_ (Δ > 4.9%)) and 31.25% (5/16) were categorized as non-responders (Δ ≤ 4.9%). There were no significant differences (all *p* < 0.05) in several potential influencing factors of responder/non-responder, including age, baseline VO_2max_, exercise adherence, and sex. In the personalized MICT + HIIT group, the incidence of individuals who experienced a favorable change in VO_2max_ was significantly (*p* < 0.05) greater when compared to the standardized MICT group. There was a positive improvement in VO_2max_ (Δ > 4.9%) in 100% (16/16) of the individuals in the personalized MICT + HIIT group.

### 3.2. Incidence of MetS z-Score Responders and Non-Responders

The incidence of MetS z-score responders and non-responders to exercise training in both the standardized MICT and personalized MICT + HIIT groups are shown in Figure 3. In the standardized MICT group, 56.25% (9/16) of individuals experienced a favorable change in MetS z-score (Δ > −0.48) and were categorized as responders. Alternatively, 43.75% (7/16) of individuals in the standardized MICT group experienced an undesirable change in MetS z-score (Δ ≤ −0.48) and were categorized as non-responders to exercise training. There were no significant differences (*p* < 0.05) in several potential influencing factors of responder/non-responder, including age, baseline MetS z-score, exercise adherence, and sex. In the personalized MICT + HIIT group, the incidence of individuals who experienced a favorable change in MetS z-score was significantly (*p* < 0.05) greater when compared to the standardized MICT group. Indeed, there was a positive improvement in MetS z-score (Δ > −0.48) in 100% (16/16) of the individuals in the personalized MICT + HIIT group.

## 4. Discussion

To the best of our knowledge, this novel study is the first prospective, randomized, controlled trial to compare individual variation in CRF and cardiometabolic health training responses following personalized MICT + HIIT versus standardized MICT, with both exercise interventions combined with a resistance training component prescribed according to recommended guidelines by two world leading professional organizations in the field of exercise science (ACSM or ACE). The major finding of the present study was that 100% of participants in the ‘personalized MICT + HIIT group’ improved CRF and cardiometabolic health, depicted as a reduction in MetS severity (MetS z-score), beyond the technical error (responders: CRF Δ > 4.9%; MetS z-score Δ > −0.48) following the 13-week exercise program. Whereas, 31.25% (5/16) and almost half (43.75%, 7/16) of the participants in the standardized MICT group showed an undesirable change in both CRF and MetS z-score following the training intervention, respectively. These results suggest that the inter-individual variability in CRF and MetS z-score responses to exercise training can be attenuated by a more personalized approach to exercise prescription. Therefore, these results support both our research hypotheses and emphasize the importance of a personalized and comprehensive approach to the exercise prescription.

The present study extends our previous findings [7,8] which also showed a higher proportion of individuals who improved CRF [7] and MetS severity [8] following an exercise intervention with an exercise intensity prescribed according to a threshold-based method (i.e., ventilatory thresholds) relative to a standardized protocol (i.e., %HRR). The method of exercise intensity prescription has been purported previously to explain the inter-individual variation in CRF training responsiveness [26]. For instance, past exercise training studies [21,26,27,28] that have employed a relative exercise intensity method (e.g., %HRR or %VO_2max_) elicited considerable inter-individual variability in metabolic and CRF responses. Alternatively, in an effort to provide a more personalized training stimulus for individuals with heterogeneous fitness levels and elicit more consistent positive training CRF responses, it has been proposed that the exercise intensity prescription should incorporate a threshold-based method [15].

The reduced variability in training response following personalized MICT + HIIT compared to MICT in the present study could also be attributed to the difference in overall exercise volume and intensity (training load). This notion is supported by studies [29] that also reported a minimal number of ‘non-responders’ with a sufficient training load. Williams et al. [29] showed that high-volume HIIT with a higher training load induced significantly more responders to CRF improvement relative to low-volume HIIT or MICT. This underpins the contention that some individuals may be ‘dose sensitive’ rather than a ‘non-responder’ [30]. Taken together, it could be inferred that a higher training load is required in individuals deemed as ‘dose sensitive’ or those considered a ‘low responder’. Moreover, it should also be acknowledged that the observed heterogeneity in VO_2max_ response to MICT might have resulted from a combination of random within-individual variation and/or technical error [31,32]. It should be noted, however, that the present study calculated a site-specific technical error through a test-retest procedure [24] by measuring both CRF and cardiometabolic parameters twice at baseline, which is considered a more robust approach than simply deriving this value from a previous study [33]. Lastly, it has also been suggested that without considering both the technical error and minimal clinically important difference (MCID), the probability of identifying a training response may be uncertain. Indeed, this may explain why the present study reflected a much greater number of ‘responders’ relative to that reported by a most recent study [29].

Another novel aspect to the present study was a comprehensive approach to the exercise prescription with the addition of a single session per week of HIIT combined with resistance training. Indeed, this personalized combination of MICT + HIIT in conjunction with resistance training proved to be a potent combination as evidenced by the 100% incidence of VO_2max_ and MetS z-score responders. The present finding is also important because the incorporation of both aerobic and resistance exercise within a training session is reflective of the usual practice of health and fitness professionals in a real-world setting. Moreover, it could be argued that a mixture of MICT + HIIT versus one or the other solely, is likely to be more sustainable in the long term. Thus, this study provides practitioners with evidence to better optimize exercise prescription incorporating aerobic and resistance training to improve cardiometabolic health. In contrast, previous research on training responsiveness has only exclusively investigated the independent effect of HIIT and MICT in the aerobic domain [29], limiting the applicability of their findings in typical practice. Moreover, this is the first study to provide a more extensive view of the cardiometabolic response (CRF and MetS severity) to a comprehensive exercise program. In support of our present findings, combined aerobic and resistance training has been shown to be more effective relative to aerobic or resistance training alone in reducing MetS severity [34].

It should also be noted that it is possible that the difference in one component of the exercise prescription (i.e., aerobic or resistance) could have induced the most influence on training responsiveness. For example, it is plausible that the mere difference in resistance training prescription (ACE [15] versus ACSM [13] recommendations) between groups could have solely accounted for the difference in training responsiveness rather than the introduction of HIIT or the personalized method of exercise intensity prescription used. Nevertheless, we could at least infer from the present study that the presence of vigorous aerobic exercise or personalized MICT coupled with resistance training may be required to enhance training responsiveness. Our results support previous findings showing superior independent effects of vigorous-intensity exercise [10,35] and combined aerobic and resistance training [6] on CRF and cardiometabolic health improvement relative to moderate-intensity exercise or only either resistance or aerobic training alone. Alternatively, we could also theorize that the accumulated sum of each component of this novel exercise prescription (personalized MICT + HIIT in conjunction with resistance training) is required to induce 100% training response. Further studies are warranted to test this hypothesis.

Ensuring that a maximal aerobic value is achieved when investigating any changes due to modifying or differing exercise doses is imperative to understand the true changes occurring from the intervention and not owing to how aerobic capacity is measured. Historically, a plateau in VO_2max_ (i.e., ≤ 150 mL·min^−1^) at the ending stages of a GXT has been the primary criterion to determine ‘true’ VO_2max_. However, there has been inconsistency regarding the value for a plateau with the use of ≤ 150 mL·min^−1^ to ≤ 50 mL·min^−1^ [36] and there has been an incidence in plateau ranging from 0% to 100% [37]. As such, VO_2peak_ rather than VO_2max_ has been recorded as a common aerobic outcome measure, especially when reporting functional capacity, fitness changes, and used to prescribe exercise interventions [38,39]. Unfortunately, VO_2peak_ does not directly indicate that aerobic adaptations have occurred due to the lack of sensitivity of this measurement. More recently, the use of a verification protocol to confirm VO_2max_ has been identified as a practical and sensitive measure to ensure that a ‘true’ maximal aerobic value has been achieved [35]. Furthermore, participants tend to exhibit greater maximal effort following a training intervention due to expecting improvements [40]. Indeed, it could be noted that a change in VO_2peak_ could be due to a greater increase in effort if there is minimal consideration of the testing methodology. With the use of a verification protocol, this overestimation of training adaptations is mitigated by ensuring all measurements are a ‘true’ VO_2max_ and representative of training adaptations.

### Limitations

Limitations to the present study merit discussion. First, although participants were advised at the beginning of the study to maintain regular dietary intake throughout the 13-week intervention, dietary intake patterns were not strictly controlled nor quantified during this investigation and may have influenced the findings. Similarly, activities of daily living and sedentary behaviors outside the intervention were not monitored; and thus could have possibly altered various outcome measurements.

## 5. Conclusions

There is robust scientific literature demonstrating an independent, dose–response relationship between improved CRF and cardiometabolic health and reduced risk of mortality from CVD and all-causes [41,42]. In the present study, a personalized exercise prescription combining MICT + HIIT in conjunction with resistance training elicited significantly greater improvements in VO_2max_ and MetS z-score reductions combined with diminished inter-individual variation in VO_2max_ and cardiometabolic training responses when compared to standardized MICT. These novel findings are encouraging and provide insightful data for the design of personalized exercise prescriptions that will optimize training responsiveness.

## Figures and Tables

**Figure 1 ijerph-16-02088-f001:**
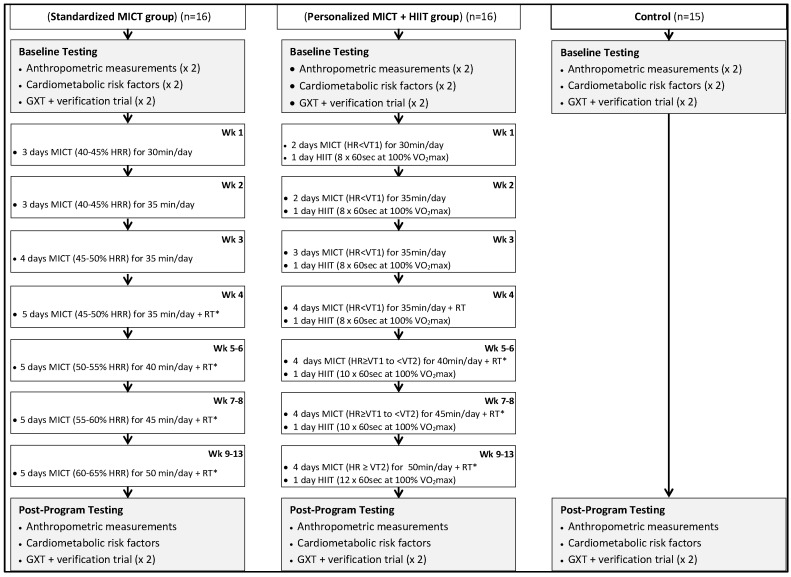
Experimental flow diagram and week-to-week exercise prescription for days/times of cardiorespiratory and resistance training. GXT = maximal graded exercise test; MICT = moderate intensity, continuous training; HIIT = high-intensity interval training; HR = heart rate; RT = resistance training; VT1 = ventilatory threshold 1; VT2 = ventilatory threshold 2; * RT performed 3 days/week.

**Figure 2 ijerph-16-02088-f002:**
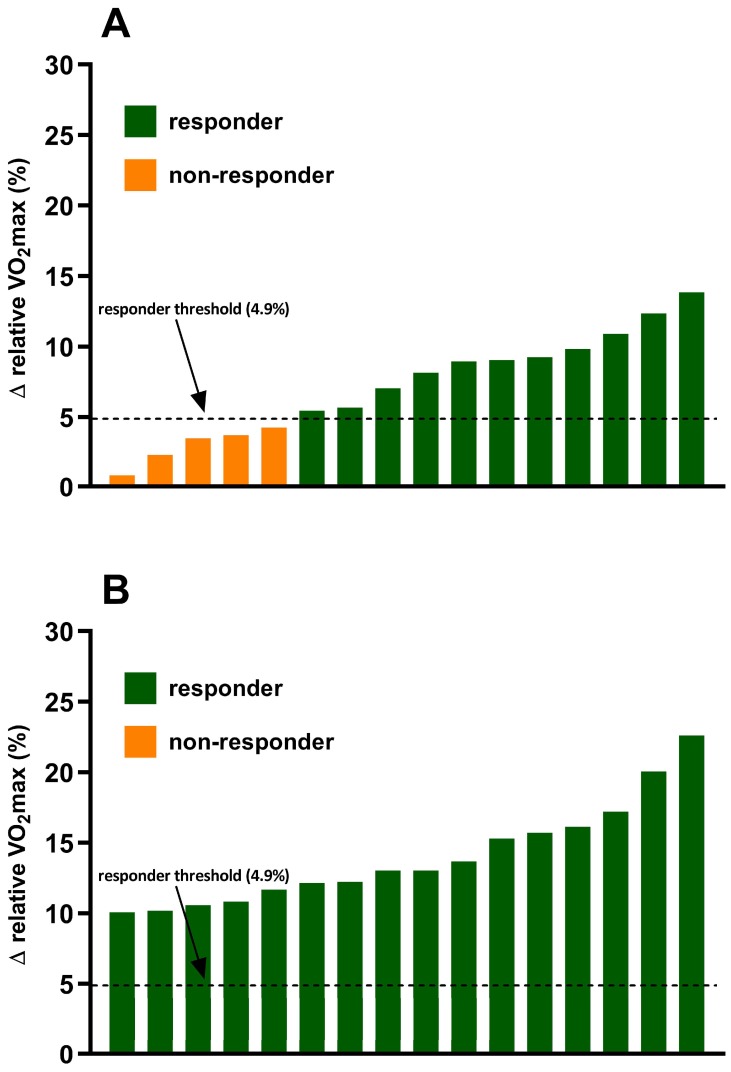
Inter-individual variability in VO_2max_ responses to exercise training in the (**A**) standardized MICT group and (**B**) personalized MICT + HIIT group.

**Figure 3 ijerph-16-02088-f003:**
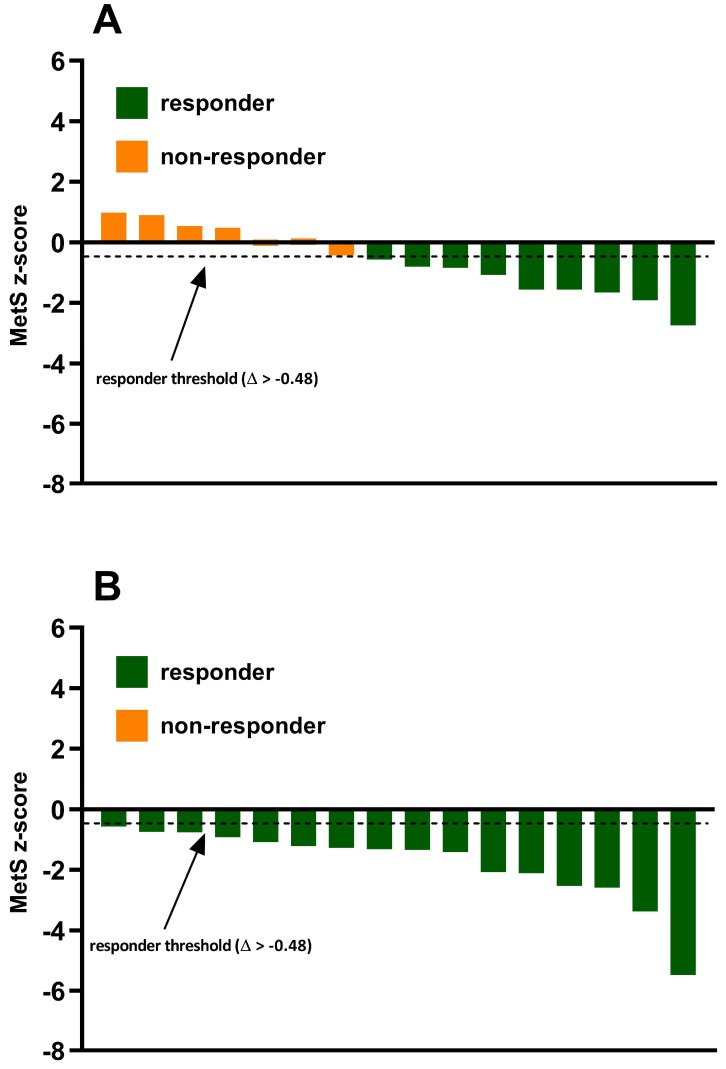
Inter-individual variability in MetS z-score responses to exercise training in the (**A**) standardized MICT group and (**B**) personalized MICT + HIIT group.

**Table 1 ijerph-16-02088-t001:** Physical and physiological characteristics at baseline and 13 weeks for control, standardized MICT, and personalized MICT + HIIT groups. (Values are mean ± SD).

Parameter	Control Group (*n* = 15; Women = 8, Men = 7)		Standardized MICT Group (*n* = 16; Women = 9, Men = 7)		Personalized MICT + HIIT Group (*n* = 16; Women = 8, Men = 8)	
	Baseline	13 Week	Baseline	13 Week	Baseline	13 Week
Age (year)	33.9 ± 6.9	____	34.2 ± 9.8	____	32.1 ± 6.9	____
Height (cm)	168.2 ± 5.9	____	167.4 ± 10.0	____	170.5 ± 8.2	____
Weight (kg)	77.3 ± 9.0	77.5 ± 8.6	80.5 ± 16.5	79.6 ± 16.0 *	79.8 ± 19.7	79.4 ± 19.6
Waist circumference (cm)	81.5 ± 7.5	81.8 ± 7.3	84.0 ± 9.3	82.8 ± 8.7 *^,†^	80.5 ± 11.9	78.8 ± 11.5 *^,†^
Body fat (%)	23.2 ± 4.9	23.8 ± 4.5 *	26.1 ± 6.2	24.6 ± 5.5 *^,†^	24.5 ± 8.5	22.1 ± 7.3 *^,†^
Resting heart rate (b∙min^−1^)	62.3 ± 6.8	63.7 ± 5.7	61.4 ± 8.9	60.3 ± 6.9	64.3 ± 5.9	62.8 ± 6.6
Bench press 5-RM (kg)	43.4 ± 13.1	43.7 ± 12.7	42.0 ± 5.7	48.1 ± 5.2 *^,†^	44.8 ± 11.8	54.3 ± 14.1 *^,‡^
Leg press 5-RM (kg)	125.2 ± 37.3	126.6 ± 34.0	124.9 ± 16.6	145.3 ± 10.4 *^,†^	121.1 ± 47.6	151.0 ± 43.8 *^,†^
VO_2max_ (mL⋅kg^−1^⋅min^−1^)	31.5 ± 6.7	30.9 ± 6.4	28.6 ± 6.0	30.8 ± 7.1 *^,†^	31.8 ± 4.8	36.3 ± 5.5 *^,‡^
Mean arterial pressure (mmHg)	91.8 ± 7.8	93.3 ± 5.4	94.1 ± 9.9	92.7 ± 6.4	92.7 ± 10.7	88.6 ± 8.3 *^,†^
HDL cholesterol (mg∙dL^−1^)	53.9 ± 21.0	52.5 ± 19.0	52.8 ± 12.4	55.6 ± 12.2	52.7 ± 13.4	57.2 ± 12.2 *^,†^
Triglycerides (mg∙dL^−1^)	109.7 ± 39.6	117.7 ± 38.5	145.1 ± 80.8	137.3 ± 71.7	138.4 ± 62.5	108.2 ± 33.2 *^,†^
Blood Glucose (mg∙dL^−1^)	88.7 ± 5.4	90.1 ± 6.7	96.1 ± 11.5	95.1 ± 9.0	97.4 ± 7.5	90.6 ± 6.1 *^,‡^
MetS z-score	−4.52 ± 3.45	−4.01 ± 3.35	−2.92 ± 2.98	−3.57 ± 2.61 *^,†^	−3.35 ± 2.87	−5.15 ± 2.34 *^,‡^

* Within-group change is significantly different from baseline, *p* < 0.05; ^†^ Change from baseline is significantly different than the control group, *p* < 0.05; ^‡^ Change from baseline is significantly different than control and standardized MICT groups, *p* < 0.05.

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
