# Peer review of "Personalized Moderate-Intensity Exercise Training Combined with High-Intensity Interval Training Enhances Training Responsiveness"

_ijerph, 2019, doi:10.3390/ijerph16122088_

Round 1

Reviewer 1 Report

Summary:

In the present study, the authors compared the effects of a “personalized” exercise training prescription (combination of moderate intensity plus high intensity) versus the “standardized” training prescription (moderate intensity) on cardiometabolic factors and cardiorespiratory fitness.  The authors report that using a “personalized” approach (based on lactate threshold) is more effective at increasing VO2max and MetS z-scores (based on fasting glucose, triglycerides, HDL, mean arterial pressure, and waist circumference) compared to a “standardized” approach (based on % HR).     

Comments, Concerns, and Suggestions:

1)      The title is quite long.  The authors should consider a shortened version.

2)      Additional data should be included.

a.       Fasting glucose values, triglycerides, HDL and mean arterial pressure are factors in the MetS z-score.  The authors should provide these data as part of Table 1, as separate figures, or a supplemental table.

b.       The authors state that resting heart rate was collected.  These data should also be included.

c.       The authors indicate that muscular fitness assessments were performed (page 5).  Were these for all subjects or those in the MICT+HIIT group to determine initial resistance training intensity?  If for all, these data should be included in the results.  If for only the “personalized” group, these methods should be included in Methods 2.6 for clarity.

3)      On page 5 (lines 7 and 9)… is the portion of the formula using TG correct?  The way it is written the calculation would be TG minus 150 divided by 69 or TG minus 2.17.

4)      On page 6, the authors use the term “site specific TE”.

a.       What does “TE” stand for?

b.       The term “site specific” needs clarification.  Were the subjects trained at different sites?  If so, how was this controlled?  Do different sites have different CVs?  If this does not refer to different study sites, perhaps, a less confusing term could be used.

5)      The authors use the term “holistically” (page 2, line 29) and “holistic” (page 11, line 1).  This term is not appropriate measures focused on cardiometabolic factors and did not include a “whole body approach” including mental, social, etc.

6)      In abstract (line 19), Minor grammar errors

a.       Purpose statement (page 2, line 27-28) ends with a question mark.

b.       Page 6, line 4 – “collectively experience” should be “collective experience”.

c.       Abstract (line 19) - “continuous” needs to be inserted between intensity and exercise.

7)      Although the electronic version of the manuscript uses different colors for Figure 2 and Figure 3, when printed in black and white, the bars are similar.  The authors might consider the use of a light and dark color for those readers that might not print the paper in color.

Author Response

Responses to reviewer #1

1)      The title is quite long.  The authors should consider a shortened version.

The title was shortened as suggested. Thank you.

2)      Additional data should be included.

a.       Fasting glucose values, triglycerides, HDL and mean arterial pressure are factors in the MetS z-score.  The authors should provide these data as part of Table 1, as separate figures, or a supplemental table.

b.       The authors state that resting heart rate was collected.  These data should also be included.

c.       The authors indicate that muscular fitness assessments were performed (page 5).  Were these for all subjects or those in the MICT+HIIT group to determine initial resistance training intensity?  If for all, these data should be included in the results.  If for only the “personalized” group, these methods should be included in Methods 2.6 for clarity.

Additional data was included in the revised manuscript as suggested for each of the above points. Thank you.

3)      On page 5 (lines 7 and 9)… is the portion of the formula using TG correct?  The way it is written the calculation would be TG minus 150 divided by 69 or TG minus 2.17.

We carefully reviewed this formula and it is correct.

4)      On page 6, the authors use the term “site specific TE”.

a.       What does “TE” stand for?

TE refers to technical error. We have spelled out technical error rather than abbreviating as it will be more clear for readers.

b.       The term “site specific” needs clarification.  Were the subjects trained at different sites?  If so, how was this controlled?  Do different sites have different CVs?  If this does not refer to different study sites, perhaps, a less confusing term could be used.

Site specific simply means we developed a unique technical error value (i.e., CVs) for our research site rather than using one sourced from the literature. Because the technical error values were developed with data from our own participants it is more valid. We added a reference to one of our earlier papers (reference 26) the provides more background on the rationale for a site specific technical error value being developed. 

5)      The authors use the term “holistically” (page 2, line 29) and “holistic” (page 11, line 1).  This term is not appropriate measures focused on cardiometabolic factors and did not include a “whole body approach” including mental, social, etc.

The terms holistically and holistic have been removed as suggested.

6)      In abstract (line 19), Minor grammar errors

a.       Purpose statement (page 2, line 27-28) ends with a question mark.

Corrected.

b.       Page 6, line 4 – “collectively experience” should be “collective experience”.

Corrected.

c.       Abstract (line 19) - “continuous” needs to be inserted between intensity and exercise.

                        Corrected.

7)      Although the electronic version of the manuscript uses different colors for Figure 2 and Figure 3, when printed in black and white, the bars are similar.  The authors might consider the use of a light and dark color for those readers that might not print the paper in color.

Excellent suggestion. We have revised the color scheme so that it clearly distinguishes between responders and nonresponders in either color or black/white print.

Reviewer 2 Report

Overview:The study by Byrd et al. examined whether the addition of one session of HIIT per week (in replace of one MICT) to a traditional exercise training intervention (MICT+Resistance) would improve training responsiveness for markers of cardiometabolic health. Interestingly, the addition of just one HIIT session per week resulted in greater improvements for VO2max and metabolic risk (Metabolic Z score). Overall the findings are of interest, results are clearly presented and the study is well done. 

Minor comments: 

Please provide a rationale for why a change of less than 5% was deemed as the threshold for a responder vs non-responder?

Should it be a clinically relevant change instead?

Were the muscular fitness assessments completed pre and post? 

What advice were the control group given -was it just to continue everyday living? 

Author Response

Responses to reviewer #2

Please provide a rationale for why a change of less than 5% was deemed as the threshold for a responder vs non-responder?

Should it be a clinically relevant change instead?

In our original version of the statistical analyses section (paragraphs 3 and 4), we explained the procedures for calculating individual responsiveness criteria for VO2max and MetS z-score. However, in response to your comment, and to add clarity for readers we added the following statements to our revised manuscript in the statistical analyses section:

End of paragraph 3 –

The calculated site-specific technical error for VO2max equated to %Δ > 4.9%.   

End of paragraph 4 –

The calculated site-specific technical error for MetS z-score equated to Δ > -0.48.

Were the muscular fitness assessments completed pre and post? 

Assessments for bench press 5-RM and leg press 5-RM were performed at baseline and 13wk. These data were added to Table 1 of our revised manuscript. Thank you.

What advice were the control group given -was it just to continue everyday living? 

Yes, this statement was added to our revised manuscript:

Participants were randomized to a non-exercise control group (who were instructed to continue their usual lifestyle habits)…